# Dimeric 1,4-Benzoquinone Derivatives with Cytotoxic Activities from the Marine-Derived Fungus *Penicillium* sp. L129

**DOI:** 10.3390/md17070383

**Published:** 2019-06-26

**Authors:** Hui-Min Zhang, Chuan-Xia Ju, Gang Li, Yong Sun, Yu Peng, Ying-Xia Li, Xiao-Ping Peng, Hong-Xiang Lou

**Affiliations:** 1Department of Natural Medicinal Chemistry and Pharmacognosy, School of Pharmacy, Qingdao University, Qingdao 266021, China; 2Key Laboratory of Chemical Biology of Ministry of Education, Department of Natural Product Chemistry, School of Pharmaceutical Sciences, Shandong University, Jinan 250012, China

**Keywords:** marine-derived fungus, *Penicillium* sp., secondary metabolites, quorum sensing, cytotoxicity, antifungal activity

## Abstract

Two new dimeric 1,4-benzoquinone derivatives, peniquinone A (**1**) and peniquinone B (**2**), a new dibenzofuran penizofuran A (**3**), and a new pyrazinoquinazoline derivative quinadoline D (**4**), together with 13 known compounds (**5**–**17**), were isolated from a marine-derived fungus *Penicillium* sp. L129. Their structures, including absolute configurations, were elucidated by extensive spectroscopic data and electronic circular dichroism calculations. Compound **1** exhibited cytotoxicity against the MCF-7, U87 and PC3 cell lines with IC_50_ values of 12.39 µM, 9.01 µM and 14.59 µM, respectively, while compound **2** displayed relatively weak cytotoxicity activities against MCF-7, U87 and PC3 cell lines with IC_50_ values of 25.32 µM, 13.45 µM and 19.93 µM, respectively. Furthermore, compound **2** showed weak quorum sensing inhibitory activity against *Chromobacterium violaceum* CV026 with an MIC value of 20 μg/well.

## 1. Introduction

Marine-derived fungi are an important resource of structurally novel and unique substances. These secondary metabolites often possess diverse biological activities, enabling producers to adapt to extremely challenging environments and contributing to the complex association with other organisms in certain ecological niches [1,2]. Enthusiasm for biologically active and novel molecules has presented opportunities in the realm of marine fungi-derived secondary metabolites. A novel diketopiperazine was characterized from the mangrove-derived *Penicillium brocae*. It exhibited potent cytotoxicity against both sensitive and cisplatin-resistant human ovarian cancer cells and established strong antimicrobial activity against pathogenic *Staphylococcus aureus* [3]. Two novel lipopeptidyl benzophenones from the fungus *Aspergillus* sp. collected from marine-submerged decaying wood displayed significant cytotoxicity on diverse cancer cells [4]. A novel dimeric nitrophenyl trans-epoxyamide was obtained from the deep-sea-derived fungus *Penicillium chrysogenum* and showed an inhibitory effect on the production of the proinflammatory cytokine IL-17 [5]. Marine-derived fungi hold great potential for the discovery of new bioactive natural products (NPs).

During our search for bioactive metabolites from marine-derived fungi, *Penicillium* sp. L129 was isolated from the rhizosphere-soil of *Limonium sinense* ( Girald ) Kuntze collected in Yangkou Beach in Qingdao, China, and was rationally selected for chemical investigation after chemical and biological evaluation of its crude extract. *Penicillium* is a productive fungal genus and has been reported to produce many new NPs, such as citrinin dimer derivatives [6,7], quinazoline [8,9] and indole diterpenoids [10]. Extensive chromatographic separation of the EtOAc extract of the fermented cultures resulted in the isolation of two new dimeric 1,4-benzoquinone derivatives, peniquinone A (**1**) and peniquinone B (**2**), a new dibenzofuran derivative, penizofuran A (**3**) and a new pyrazinoquinazoline derivative quinadoline D (**4**), along with 13 known compounds, i.e., quinadoline A (**5**) [11], 3,4-dimethoxy-5-methylphenol (**6**) [12], orcinol (**7**) [13], 1,3,5,6-tetrahydroxy-8-methylxanthone (**8**) [14], mucorisocoumarins A (**9**) [15], penicillic acid (**10**) [16], dihydropenicillic acid (**11**) [16], dechlorogriseofulvin (**12**) [17], griseofulvin (**13**) [18], isogriseofulvin (**14**) [19], dehydrogriseofulvin (**15**) [20], *trans*-capsaicin (**16**) [21], and dihydrocapsaicin (**17**) [22] (Figure 1). Their structures were elucidated through several spectroscopic methods. Herein we report the structures and bioactivities of these compounds.

## 2. Results and Discussion

### 2.1. Structural Elucidation

Peniquinone A (**1**) was obtained as a red oil. Its molecular formula was deduced as C_17_H_18_O_6_ with nine degrees of unsaturation according to the HRESIMS peak at *m/z* 319.1171 [M + H]^+^ (Appendix A). Analysis of the combined 1D NMR spectral data (Table 1) established that **1** possessed two aromatic methyl groups (6-CH_3_ and 6’-CH_3_), three methoxy groups (2-OCH_3_, 4′-OCH_3_ and 5′-OCH_3_), two aromatic protons (C-3 and C-3’) as well as a benzene ring. The connection mode and substitution pattern on the 1,4-benzoquinone and benzene rings were determined by key HMBC and NOSEY correlations (Figure 2). Key HMBC connections of 2-OCH_3_/C-2, 4’-OCH_3_/C-4’, 5’-OCH_3_/C-5’, 6’-CH_3_/C-1’, 6’-CH_3_/C-5’, and 6’-CH_3_/C-6’ and the chemical shift of C-2’ established that the phenol fragment of compound **1** was 3,4-dimethoxy-5-methylphenol (**6**) [12]. The 1,4-benzoquinone ring system was also confirmed by HMBC and NOESY correlations, which was further supported by comparing its NMR data with those of reported derivatives [23,24]. Compound **1**, as a new compound containing a dimeric 1,4-benzoquinone motif and a phenol fragment, was thus elucidated as 2’-hydroxy-2,4’,5’-trimethoxy-6,6’-dimethyl-(5,1’-biphenyl)-1,4-dione.

Peniquinone B (**2**) was isolated as a red oil. The molecular formula was determined to be C_15_H_14_O_5_ on the basis of HRESIMS *m/z* 273.0767 [M − H]^−^ (Appendix A), along with the ^1^H and ^13^C NMR data (Table 1), indicating nine degrees of unsaturation. Its 1D NMR spectra were similar to those of **1**, except for an additional hydroxyl signal and an aromatic proton (Table 1, Appendix A). In addition, two aromatic proton signals exhibited a small coupling constant (*J* = 1.9 Hz) (Table 1). The above analysis indicated that there was resorcinol substructure in compound **2**, which was further supported by HMBC correlations (Figure 2, Appendix A). Therefore, compound **2** was constructed by a 1,4-benzoquinone moiety linking with a resorcinol group. The resorcinol fragment of compound **2** was orcinol (**7**) [13], which was also a known structure. Thus, compound **2** was elucidated as 2’,4’-dihydroxy-2-methoxy-6,6’-dimethyl-(5,1’-biphenyl)-1,4-dione.

Penizofuran A (**3**) was obtained as a brown powder, and its molecular formula C_16_H_16_O_5_ was established from the HRESIMS *m/z* 289.1079 [M + H]^+^ (Appendix A). Nine degrees of unsaturation were indicated besides the ^1^H and ^13^C NMR data (Table 2). The ^1^H NMR spectrum (Table 2) of **3** showed signals corresponding to one methyl (1-CH_3_), three methoxys (3-OCH_3_, 7-OCH_3_ and 8-OCH_3_) and three aromatic protons (C-4, C-6 and C-9). The ^13^C NMR spectrum showed sixteen carbon signals including twelve aromatic, one methyl, and three methoxyl groups. As two aromatic rings accounted for eight degrees of unsaturation, compound **3** was inferred to have an additional ring. Therefore, it was assumed that **3** has a dibenzofuran skeleton [25]. The position of the methyl, methoxy and hydroxyl in the dibenzofuran nucleus was established by 2D NMR data (Figure 2, Appendix A), and the key HMBC connections of 3-OCH_3_/C-3, 7-OCH_3_/C-7, 8-OCH_3_/C-8, 1-CH_3_/C-1, 1-CH_3_/C-2, and 1-CH_3_/C-9b, and NOESY connections between 3-OCH_3_ and C-4, 7-OCH_3_ and C-6, 8-OCH_3_ and C-9. Compound **3**, as a new compound containing the parent nucleus of dibenzofuran, was thus elucidated as 3,7,8-trimethoxy-1-methyldibenzo[*b,d*]furan-2-ol. 

Quinadoline D (**4**) was obtained as a yellow powder. Its molecular formula was determined as C_27_H_25_N_5_O_4_ according to the HRESIMS peak at *m/z* 484.1966 [M + H]^+^ (Appendix A). The IR and UV spectra of **4** were very similar to those of quinadoline A (**5**) [11]. This suggested that compound **4** was a pyrazinoquinazoline derivative. Comparing its MS and NMR data with those in compound **5** (Table 3, Appendix A), compound **4** has 2 amu less than **5** in the MS spectrum, and it showed vicinal-coupling methylene groups (31-CH_2_ and 32-CH_2_) in **4**, instead of two methyl groups in **5**. The above cyclopropane fragment was further observed in the COSY spectrum (Figure 2), and verified by the key HMBC correlations from both H-31 and H-32 to C-22 and C-23. The NOE correlation observed between H-20 and H-15a, and absence of an NOE correlation between H-20 and OH-19 (Appendix A) revealed that H-20 and OH-19 were cofacial. The ECD spectrum of compound **4** was matched to **5** (Figure 3). It was hypothesized that compound **4** has the same absolute configuration as that of **5**. ECD calculations were further employed to determine the absolute configuration. The predicted ECD spectrum was obtained by the TDDFT [mPW1PW91/6-311G(d)] method, and was subsequently compared with the experimental data (Appendix A). As shown in Figure 3, the calculated ECD spectrum was in good accordance with the experimental curve, confirming the absolute configuration of compound **4** as 14*R*, 19*S*, 20*R* (Figure 3).

### 2.2. The Bioactivities of Compounds 

Compounds **1**–**4** were tested for their cytotoxicity against MCF-7, A549, U87 and PC3 cancer cell lines using the 3-(4,5)-dimethylthiahiazo(-z-y1)-3,5-di-phenytetrazoliumromide (MTT) method [26] (Table 4). Adriamycin was employed as the positive control in the cytotoxicity test. Compound **1** showed cytotoxicity against the MCF-7, U87 and PC3 cell lines with IC_50_ values of 12.39 µM, 9.01 µM and 14.59 µM, respectively. Compound **2** also exhibited cytotoxicity against MCF-7, U87 and PC3 cell lines with IC_50_ values of 25.32 µM, 13.45 µM and 19.93 µM, respectively. Compounds **1**–**4** were also evaluated for their antibacterial activities against the bacteria *Staphylococcus aureus*, *Bacillus subtilis*, *Pseudomonas aeruginosa* and *Escherichia coli*, by applying the disk diffusion method, according to the Clinical and Laboratory Standards Institute (CLSI) [27]. Unfortunately, none of these compounds were effective against the tested microorganisms.

All of the compounds were evaluated for their QS inhibitory activity against *C. violaceum* CV026. C30 was applied as the positive control in the QS inhibitory activity test. Compounds **2** and **8** displayed weak QS inhibitory activity against *C. violaceum* CV026 both with an MIC value of 20 μg/well (Appendix A). Compound **10** was a classical QS inhibitory active compound, penicillic acid. By comparing compounds **10** and **11** with structurally related metabolites, it was revealed that alterations to the alkene bond side chain substitutions could significantly influence their QS inhibitory activity. In addition, all of the compounds were tested for their antifungal activities against four agricultural pathogens *Colletotrichum musae* (ACCC 31244), *Colletotrichum coccodes* (ACCC 36067), *Colletotrichum asianum* T0408 and *Magnaporthe grisea* (ACCC 37631), using the disk diffusion method according to the CLSI [27,28]. The results showed that compounds **12**–**15** had potent inhibitory activities against *C. musae* (ACCC 31244), and compound **14** also showed inhibitory activity against *C. coccodes* (ACCC 36067) (Table 5 and Appendix A). The positive control used in the antifungal activity test was cycloheximide.

## 3. Materials and Methods

### 3.1. General Experimental Procedures

Optical rotations were measured with a JASCO P-1020 digital polarimeter (JASCO Corporation, Tokyo, Japan). ECD spectra were measured on a JASCO J-715 (JASCO) or Chirascan CD (Applied Photophysics) spectropolarimeter. IR spectra were recorded on a Nicolet iS5 (Thermo Fisher Scientific, USA) in KBr discs. NMR spectra were obtained on a Bruker Advance spectrometer operating at 500 (^1^H) and 125 (^13^C) MHz using TMS as an internal standard, with chemical shifts recorded as *δ* values. Mass spectra were obtained on an LTQ-Orbitrap spectrometer equipped with an ESI source. Flash chromatography was performed on a Teledyne ISCO CombiFlash Rf 200 system. The high performance liquid chromatography (HPLC) system (Tianjin Bonna-Agela Technologies Co., Ltd., China) was composed of two HP-Q-P050 high pressure pumps equipped with a HP-Q-UV100S variable UV detector, a ATS-051-H10 automatic sampler, a FL-C100B fraction collector, and a Innoval column (10 × 250 mm, 10μm). The semi-preparative HPLC system (Agilent 1260 Infinity II; Agilent technologies, Germany) was equipped with a 1260 Quat Pump VL, a 1260 Vialsampler, a 1260 MCT, a 1260 DAD *WR*, and an ZORBAX SB-C18 column (5 μm, 9.4 × 250 mm). Thin-layer chromatography (TLC) was performed with silica gel GF_254_ plates (Qingdao Haiyang Chemical Co., Ltd., China). Column chromatography was performed on silica gel (200–300 mesh, Qingdao Haiyang Chemical Co., Ltd., China). Size exclusion chromatography was performed using Sephadex LH-20 (25–100 μm; Pharmacia, Uppsala, Sweden).

### 3.2. Fungal and Plant Material

The fungal strain *Penicillium* sp. L129 was isolated from the rhizosphere-soil of *Limonium sinense* (Girald) Kuntze collected in Yangkou Beach, Qingdao, China. It was identified according to its morphological characteristics and the analysis of the internal transcribed spacer (ITS) region of 16S rDNA (Genbank access No. MK625482). The fungus was deposited at the School of Pharmacy, Qingdao University, China, and was maintained at −80 °C.

### 3.3. Fermentation and Extraction

For large-scale fermentation, the fresh mycelia of *Penicillium* sp. L129 were cultured on potato dextrose agar (PDA) medium at 28 ± 2 °C for five days. The agar plugs were cut into small pieces under aseptic conditions, and 150 pieces were used to inoculate 50 flasks (500 mL) each containing 80 g rice, 120 mL seawater, and 0.24 g peptone. The cultures were inoculated at 28 ± 2 °C for 40 days. After 40 days of cultivation, the fermented cultures were extracted three times with ethyl acetate (EtOAc). The organic solvent was evaporated under reduced pressure to afford the crude extract (40.7 g). 

### 3.4. Purification and Identification

The obtained EtOAc crude extract (40.7 g) based on TLC analysis was fractionated into six fractions (Fr.1 to Fr.6) by column chromatography on silica gel, eluting with a gradient of CH_2_Cl_2_–MeOH (100–50%). Fr.2 (24.5 g) was fractionated by silica gel column chromatography with a gradient of EtOAc–petroleum ether (3%–100%) to give eight subfractions (Fr.2.1 to Fr.2.8). Fr.2.2 (283.0 mg) was purified by HPLC (MeOH/H_2_O, 50/50, 4 mL/min) to yield **1** (2.0 mg, *t*_R_ 14.5 min), **6** (7.0 mg, *t*_R_ 8.3 min), and **9** (1.1 mg, *t*_R_ 22.2 min). Fr.2.3 (1.9 g) was separated through a Sephadex LH-20 chromatograph (MeOH as eluent) to give Fr.2.3.1 and Fr.2.3.2. Fr.2.3.1 (1.4 g) was separated by a CombiFlash Rf 200 purification system (eluting with MeOH–H_2_O, 5% MeOH for 5 min, a gradient of 5–100% MeOH over 20 min, 100% MeOH for 10 min), which obtained four subfractions (Fr.2.3.1.1 to Fr.2.3.1.4), and Fr.2.3.1.4 (21.8 mg) was further purified by semi-preparative HPLC (MeOH/H_2_O, 70/30, 2 mL/min) to give compounds **3** (1.7 mg, *t_R_* 14.6 min), **16** (0.8 mg, *t*_R_ 19.2 min), and **17** (0.6 mg, *t*_R_ 26.7 min). Fr.2.3.2 (64.6 mg) was applied to Sephadex LH-20 (MeOH as eluent) to yield four subfractions (Fr.2.3.2.1 to Fr.2.3.2.4), and Fr.2.3.2.2 (12.0 mg) was further purified by semi-preparative HPLC (MeOH/H_2_O, 55/45, 2 mL/min) to give compound **2** (4.8 mg, *t*_R_ 9.3 min), while Fr.2.3.2.3 (25.6 mg) was further purified by semi-preparative HPLC (MeOH/H_2_O, 50/50, 2 mL/min) to give compound **7** (11.5 mg, *t*_R_ 8.0 min). Part of Fr.2.5 (250.0 mg) was separated through a Sephadex LH-20 chromatograph (MeOH as eluent) to give Fr.2.5.1 to Fr.2.5.2, and Fr.2.5.1 (200.0 mg) was purified through HPLC (MeOH/H_2_O, 40/60, 4 mL/min) to obtain compounds **10** (150.0 mg, *t*_R_ 7.5 min) and **11** (6.0 mg, *t*_R_ 10.5 min). Fr.2.7 (128.2 mg) was purified through HPLC (MeOH/H_2_O, 55/45, 4 mL/min) to provide compounds **12** (4.3 mg, *t*_R_ 14.7 min), **13** (3.7 mg, *t*_R_ 22.9 min), **14** (0.6 mg, *t*_R_ 26.0 min), and **15** (0.9 mg, *t*_R_ 20.8 min). Fr.2.8 (0.5 g) was subjected to semi-preparative HPLC (MeOH/H_2_O, 65/35, 2 mL/min) to yield compounds **4** (1.7 mg, *t*_R_ 17.6 min) and **5** (2.2 mg, *t*_R_ 20.5 min). Then, Fr.3 (3.7 g) was fractionated by a CombiFlash Rf 200 purification system (eluting with MeOH–H_2_O, 5% MeOH for 5 min, a gradient of 5–100% MeOH over 20 min, 100% MeOH for 10 min) to give four subfractions (Fr.3.1 to Fr.3.4), and Fr.3.4 (700 mg) was subjected to semi-preparative HPLC (MeOH/H_2_O, 60/40, 2 mL/min) to yield compound **8** (9.2 mg, *t*_R_ 18.7 min).

Compound **1**. Red oil; UV (CH_3_OH) *λ_max_* (log ε) 200, 274, 352 nm; IR (KBr) *ν_max_* 3431, 2927, 1682, 1617, 1229, 1040 cm^−1^; HRESIMS *m/z* 319.1171 ([M + H] ^+^, calcd. for C_17_H_18_O_6_ 319.1176); ^13^C NMR and ^1^H NMR (Table 1).

Compound **2**. Red oil; UV (CH_3_OH) *λ*_max_ (log ε) 216, 270, 380 nm; IR (KBr) *ν*_max_ 3431, 2921, 2850, 1621, 1463, 1377, 1231, 720 cm^−1^; HRESIMS *m/z* 273.0767 ([M − H]^−^, calcd. for C_15_H_13_O_5_ 273.0768); ^13^C NMR and ^1^H NMR (Table 1).

Compound **3**. Brown powder; UV (CH_3_OH) *λ*_max_ (log ε) 226, 252, 316 nm; IR (KBr) *ν*_max_ 3438, 2922, 2852, 1632, 1463, 1384, 1261, 1119, 585 cm^−1^; HRESIMS *m/z* 289.1079 ([M + H]^+^, calcd. for C_16_H_17_O_5_ 289.1071); ^13^C NMR and ^1^H NMR (Table 2).

Compound **4**. Yellow powder; UV (CH_3_OH) *λ*_max_ (log ε) 210, 230, 292 nm; IR (KBr) *ν*_max_ 3432, 2923, 2853, 1628, 1464, 1378, 1128 cm^−1^; ${\left[\alpha \right]}_{D}^{25}$ −6. (c 0.1, MeOH); HRESIMS *m/z* 484.1966 ([M + H]^+^, calcd. for C_27_H_26_N_5_O_4_ 484.1979); ^13^C NMR and ^1^H NMR (Table 3).

### 3.5. QS inhibitory Activity Essay 

The in vitro QS inhibitory activity assay was carried out against the *C. violaceum* CV026. The strain *C. violaceum* CV026 was inoculated in a 20 mL LB broth medium (NaCl 10 g/L, Tryptone 10 g/L, Yeast Extract 5 g/L) at 37 °C on a rotary shaker (220 rpm) overnight. The culture (0.2 mL) was further mixed with 15 mL of warm molten LB agar (~40 °C) containing kanamycin (Sigma, 0.72mg) and N-hexanoyl-L-homoserine-lactone (C6-HSL) (Sigma, 1.5 μg). The agar was poured into a Petri dish and then punched with a sterile cork borer (Φ 5mm). The methanol solution (10 μL) of tested compounds at the concentration of 40 μg/well were pipetted into each well. Furanone C30 at the concentration of 10 μg/well was employed as the positive control while methanol was used as the negative control. The plates were then incubated overnight at 37 °C.

### 3.6. Cytotoxic Activity Essay

The MCF-7 and PC3 cell lines were obtained from the American Type Culture Collection (Manassas, VA) [29]. The A549 cell line was purchased from the Shanghai Institute for Biological Sciences, China Academy of Sciences (China) [30]. The U87 cell line was obtained from Dr. Bing Yan (Shan Dong University) [31].

## 4. Conclusions

We identified 17 compounds from the fermentation of *Penicillium* sp. L129, including two new dimeric 1,4-benzoquinone, peniquinone A (**1**) and peniquinone B (**2**), a new dibenzofuran derivative penizofuran A (**3**) and a new pyrazinoquinazoline derivative quinadoline D (**4**). Compounds **1–4** were screened for their cytotoxic activity against MCF-7, A549, U87 and PC3 cancer cell lines, and antibacterial activities against Gram-positive bacteria *S. aureus* and *B. subtilis* and Gram-negative bacteria *P. aeruginosa* and *E. coli*. All the compounds were tested for their QS inhibitory activity against *C. violaceum* CV026 and antifungal activities against *C. musae* (ACCC 31244), *C. coccodes* (ACCC 36067), *M. grisea* (ACCC 37631) and *C. asianum* T0408. Compound **1** showed cytotoxicity against the MCF-7, U87 and PC3 cell lines with IC_50_ values of 12.39 µM, 9.01 µM and 14.59 µM, respectively, while compound **2** exhibited cytotoxicity against MCF-7, U87 and PC3 cell lines with IC_50_ values of 25.32 µM, 13.45 µM and 19.93 µM, respectively. The known compounds **13** and **14** had potent antifungal activities against *C. musae* (ACCC 31244), both with an MIC values of 0.1 µg/scrip, and compound **14** also showed antifungal activity against *C. coccodes* (ACCC 36067) with an MIC value of 0.1 µg/scrip. Compounds **2** and **8** displayed weak QS inhibitory activities against *C. violaceum* CV026 both with an MIC value of 20µg/well. Compound **10** was penicillic acid, showing obvious QS inhibitory activity. The results suggested that the marine-derived fungi are an important source of new bioactive substances.

## Figures and Tables

**Figure 1 marinedrugs-17-00383-f001:**
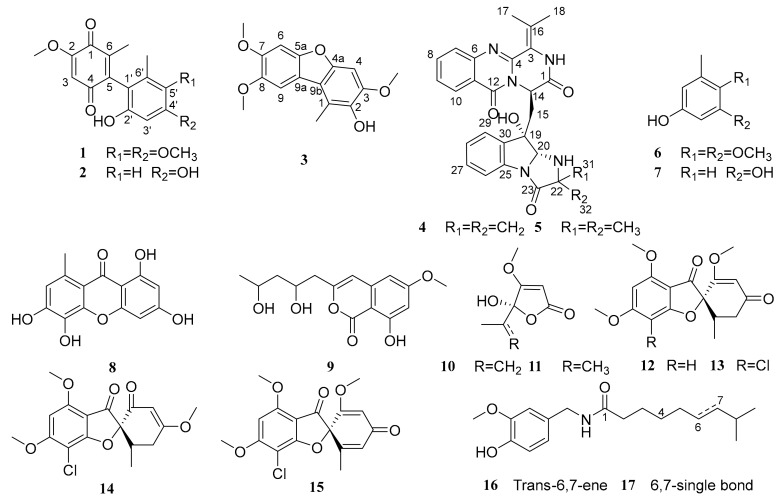
Chemical structures of compounds **1**–**17**.

**Figure 2 marinedrugs-17-00383-f002:**
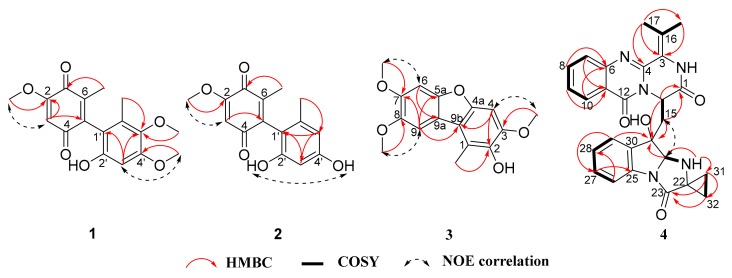
The key 2D NMR correlations for compounds **1**–**4**.

**Figure 3 marinedrugs-17-00383-f003:**
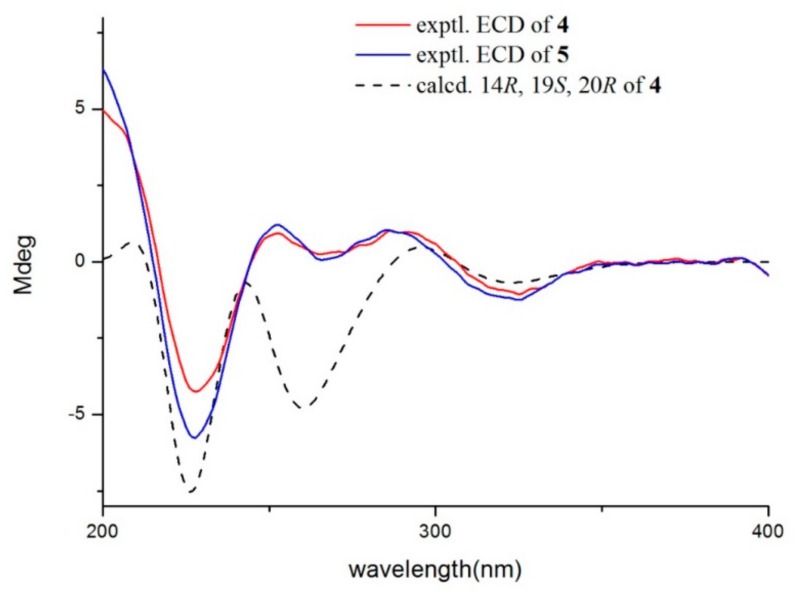
Comparison of measured and calculated ECD spectrums of **4** and **5** (Calcd. ECD curves with a bathochromic shift for 10 nm).

**Table 1 marinedrugs-17-00383-t001:** ^1^H and ^13^C NMR Data for **1** and **2** (^1^H 500 MHz, ^13^C 125 MHz, TMS, *δ* ppm).

Position	1 ^a^		2 ^b^	
	*δ*_C_, Type	*δ*_H_ (*J* in Hz)	*δ*_C_, Type	*δ*_H_ (*J* in Hz)
1	182.6, C		182.3, C	
2	158.8, C		158.3, C	
3	107.7, CH	6.01, s	107.6, CH	6.11, s
4	186.5, C		186.1, C	
5	140.9, C		141.7, C	
6	143.0, C		140.7, C	
1′	112.7, C		111.7, C	
2′	148.7, C		155.1, C	
3′	98.8, CH	6.36, s	99.9, CH	6.16, d (1.9)
4′	153.7, C		157.9, C	
5′	141.6, C		107.8, CH	6.13, d (1.9)
6′	131.0, C		137.3, C	
6-CH_3_	13.6, CH_3_	1.88, s	13.0, CH_3_	1.69, s
6′-CH_3_	13.4, CH_3_	1.96, s	19.5, CH_3_	1.82, s
2-OCH_3_	56.4, OCH_3_	3.85, s	56.3, OCH_3_	3.80, s
4′-OCH_3_	55.8, OCH_3_	3.83, s		
5′-OCH_3_	60.6, OCH_3_	3.75, s		
2′-OH				9.06, s
4′-OH				9.24, s

^a^ Recorded in CDCl_3_; ^b^ Recorded in DMSO-*d_6_*.

**Table 2 marinedrugs-17-00383-t002:** ^1^H and ^13^C NMR Data for **3** (^1^H 500 MHz, ^13^C 125 MHz, TMS, *δ* ppm).

Position	3^a^	
	*δ*_C_, Type	*δ*_H_ (*J* in Hz)
1	116.6, C	
2	140.1, C	
3	147.0, C	
4	93.1, CH	7.13, s
4a	149.1, C	
5a	150.3, C	
6	96.2, CH	7.30, s
7	148.1, C	
8	145.5, C	
9	104.1, CH	7.45, s
9a	116.3, C	
9b	115.5, C	
1-CH_3_	12.3, CH_3_	2.56, s
3-OCH_3_	56.2, OCH_3_	3.87, s
7-OCH_3_	56.0, OCH_3_	3.84, s
8-OCH_3_	56.3, OCH_3_	3.86, s

^a^ Recorded in DMSO-*d_6_.*

**Table 3 marinedrugs-17-00383-t003:** ^1^H and ^13^C NMR Data for **4** (^1^H 500 MHz, ^13^C 125 MHz, TMS, *δ* ppm).

Position	4^a^	
	*δ*_C_, Type	*δ*_H_ (*J* in Hz)
1	166.5, C	
3	121.5, C	
4	146.8, C	
6	146.8, C	
7	127.1, CH	7.66, d (8.2)
8	134.6, CH	7.85, t (7.7)
9	126.9, CH	7.55, t (7.4)
10	126.3, CH	8.16, d (8.3)
11	119.6, C	
12	159.9, C	
14	52.5, CH	5.43, t (6.5)
15	37.5, CH_2_	15a: 2.48, dd (8.2, 14.8) 15b: 2.57, dd (5.4, 14.8)
16	130.8, C	
17	21.5, CH_3_	2.31, s
18	21.1, CH_3_	1.96, s
19	75.1, C	
20	80.5, CH	5.25, d (9.5)
22	46.6, C	
23	173.5, C	
25	138.3, C	
26	124.3, CH	7.10, m
27	129.7, CH	7.33, m
28	114.4, CH	7.33, m
29	124.6, CH	7.35, m
30	137.1, C	
31	14.3, CH_2_	0.84, m
32	10.9, CH_2_	1.01, m; 0.93, m
19-OH		5.68, br s
2-NH		10.04, br s
21-NH		3.57, d (9.4)

^a^ Recorded in DMSO-*d_6_*.

**Table 4 marinedrugs-17-00383-t004:** Cytotoxic activities of compounds **1**–**4.**

Compounds	IC_50_			
	MCF-7	A549	U87	PC3
**1**	12.39 ± 2.43	>40	9.01 ± 2.36	14.59 ± 2.75
**2**	25.32 ± 3.22	>40	13.45 ± 3.12	19.93 ± 3.48
**3**	>40	>40	>40	>40
**4**	>40	>40	>40	>40
Adriamycin	2.03 ± 0.42	1.53 ± 0.37	1.21 ± 0.50	0.98 ± 0.23

**Table 5 marinedrugs-17-00383-t005:** Minimum inhibitory concentrations (MIC) of the compounds **12**–**15** against fungal pathogens^a^.

Organism	12	13	14	15
*Colletotrichum musae* (ACCC 31244)	1	0.1	<10	>1
*Colletotrichum coccodes* (ACCC 36067)	NA	NA	<10	NA

^a^ All values are in μg/scrip and derived from experiments in triplicate; NA: no activity at the concentration of 10 µg/scrip; Compound **14** did not have enough mass to get MIC data.

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
