# Peer review of "Dimeric 1,4-Benzoquinone Derivatives with Cytotoxic Activities from the Marine-Derived Fungus Penicillium sp. L129"

_marinedrugs, 2019, doi:10.3390/md17070383_

Reviewer 1 Report

Please find my comments about the results and the presentation below:

·         Table 3: Do the C8 and C9 protons really give a triplet? Maybe there is a dd?

·         Table 3: Why does not at least one C15 proton coupled to the C14 proton?

·         Table 5: The name of pathogens should be written with capital letter.

Reviewer 2 Report

Two new dimeric 1,4-benzoquinone derivatives, peniquinone A and peniquinone B  a new dibenzofuran penizofuran A  and a new pyrazinoquinazoline derivative quinadoline known  were isolated from a marine-derived fungus 16 Penicillium sp. L129 and characterized. This work is very good and should be published.

However there are few things to add - Figure1 proper description should be given for compound number 17

HPLC data for the compounds isloated should be incorporated in the SI
